# Sustainable Urban Development in Slum Areas in the City of Rio de Janeiro Based on LEED-ND Indicators

**Ana Carolina Hyczy de Siqueira** [1] , **Mohammad K. Najjar** [2] , **Ahmed W. A. Hammad** [3] , **Assed Haddad** [4] **and Elaine Vazquez** [1,*]

1   Programa de Engenharia Urbana, Universidade Federal do Rio de Janeiro, Rio de Janeiro 21941-909, Brazil; ana.hyczy@poli.ufrj.br
2   Centro Universitário Gama e Souza (UNIGAMA), Rio de Janeiro 22621-090, Brazil; mnajjar@poli.ufrj.br
3   Faculty of Built Environment, University of New South Wales, Sydney 2052, Australia; a.hammad@unsw.edu.au
4   Programa de Engenharia Ambiental, Universidade Federal do Rio de Janeiro, Rio de Janeiro 21941-909, Brazil; assed@poli.ufrj.br
*   Correspondence: elaine@poli.ufrj.br

**Abstract:** The accelerated urban transition and its consequent unsustainability is a problem registered in several global scenarios. This issue has been identified in the city of Rio de Janeiro in Brazil. One of the solutions provided for this theme is the application of specific methodologies to measure sustainability in urban areas such as the case of Leadership in Energy and Environmental Design for Neighborhood Development (LEED-ND). This work aims to analyze a real urban project, still in the executive project phase, Morro do Encontro project, in the scenario of the slum area of Rio de Janeiro based on the LEED-ND indicator system. The novelty of this study is to evaluate the existing relationships, between urban development actions and sustainability, through LEED-ND indicators, in the proposals of the Brazilian government plan PAC2, as a means of verifying their effectiveness. A total of 12 boards of the discipline of Urbanism in the executive project phase are studied. The analyzed items are divided into credit items and mandatory items. A total of 8% of credit items (CI) and 33% of mandatory items (MI) are attended. The results found indicated 47 sustainability items were not met and, therefore, can be improved. This comes back to the fact that 100% of the mandatory LEED-ND items were not achieved, which means that this project as it was conceived cannot be considered as a sustainable urban solution.

**Keywords:** urban development; growth acceleration program; sustainability indicators

## 1. Introduction

The global scenario highlights the accelerated urban transition with unsustainable consequences [1,2]. Brazil stood out in the second half of the twentieth century with one of the most accelerated urban transitions in the world, which transformed it from rural and agricultural to an urban and metropolitan country [3,4]. This vision contributed to several urban difficulties that reflect the confluence of two main factors; the inequalities and the inability to anticipate, predict, accept, and plan urban growth [5,6]. The city of Rio de Janeiro is an example that shows the extreme limits of urban inequalities resulting from these transformations. Social inequalities, social disorganization, crime, and slums are part of the urban characteristics of this city [7]. Despite the problem of Rio's slums dating back to colonial Brazil [8], the lack of land supply for housing for the low-income Brazilian population demonstrates the inability of good city planning [9]. Still unsolved, this type of urban space continues to be reproduced and growing, currently covering a number of about 763 slums in the capital of Rio de Janeiro [10].

Urban development can be defined as a contribution to the improvement of conditions, material, and subjective, of life in cities [11]. Hence, it is necessary to reduce social inequalities and guarantee social, environmental, and economic sustainability [12,13]. The lack of urban development is reflected in the housing deficit and precarious housing, in addition to the low quality of public transport and access to basic sanitation [14,15], taking into consideration that the practice of urban development includes a search for guaranteeing the rights to the city [16]. This step must be preceded by planning, given the success of its application, which could empower the decision-making process towards achieving economic, social, cultural, and environmental goals with political principles, tools, institutional and participation mechanisms, and regulatory procedures [17]. In Brazil, the milestones of urban development came through different government proposals presented in the table below. In the literature, there have been several attempts to plan and develop the Brazilian urban environment, from the creation of government agencies, banks, ministries, and statutes, to the application of national development plans and policies, as illustrated in Table 1.

**Table 1.** Landmarks of urban development in Brazil.

| Year | Name of the Event and/or Document | Source |
|---|---|---|
| 1953 | Federal agency "Housing and Urbanism" | [17] |
| The decade of the 1960s | National Housing Bank | [18] |
| 1975 | Second National Development Plan (II NDP-1975/79) | [19] |
| The decade of the 1980s | Relevance for discussions about "marginality" (started in the 1960s and 1980s) with prominence in the national political scene | [17] |
| 1985 | Ministry of Urban Development and Environment | [20] |
| 1988 | Brazilian Federal Constitution (CF/88), which instituted the Master Plan and the Right to the Environment | [21] |
| 1990 | Ministry of Social Action | [20] |
| 1992 | Production of important documents, such as Rio Declaration on Environment and Development Declaration of Principles on the Use of Forests United Nations Convention on Biological Diversity United Nations Convention on Climate Change Global Agenda 21 | [20] |
| 1992 | Rio-ECO92 | [22] |
| 2001 | City Statute | [23] |
| 2003 | Principles and guidelines of Brazilian urban policy 1st National Conference of Cities | [20] |
| 2003 | Ministry of Cities | [20] |
| 2006 | National Urban Development Policy (NUDP) | [20] |
| 2007–2010 | 1st phase of the Growth Acceleration Program (PAC) | [24] |
| 2009 | Programa Minha Casa Minha Vida (My House My Life Program) (PMCMV)—PHASE 1 | [18] |
| 2011–2014 | 2nd phase of the Growth Acceleration Program (PAC2)/ Programa Minha Casa Minha Vida (My House My Life Program) (PMCMV)—PHASE 2 | [25] |

Sustainable urban development necessitates the involvement of the four different dimensions that are the environmental, the social, the economic, and the cultural [26]. Therefore, since the 1970s, there has been worldwide recognition of the need for a change in behavior and the choice of sustainable

actions, which have been discussed and improved over time [27]. More recently in 2016, the New Urban Agenda (NAU) was established, as a result of the Habitat III Conference [28]. This agenda considered urbanization as one of the most transformative trends of the 21st century. Therefore, it aimed to guide the urbanization policy for the next 20 years [29]. However, sustainability alone is already a very complex and multifaceted subject. Its analogy in relation to urban development also behaves in the same way.

One way to assess the level of sustainability of a given element is to apply an analysis based on sustainability indicators recognized from environmental certifications [30]. In the literature, there have been several attempts to measure the sustainability of urban development, culminating in the creation of multiple sustainability indicators [31,32]. These indicators are considered instruments that allow the visualization, measurement, and evaluation of urban characteristics [33]. Hence, specialists in different areas of knowledge have been developing ways to apply and measure urban sustainability according to specific methodologies, which resulted in international initiatives to develop environmental indicators and sustainable development [34]. The application of the sustainability indicators can be associated with obtaining environmental certifications to prove sustainability in certain aspects such as the Leadership in Energy and Environmental Design (LEED®) more specifically the LEED-ND (LEED for Neighborhood and Development) neighborhood sustainability indicator system [35,36], which will be used as a research method of this work.

*Goals and Objectives*

The LEED typology for Neighborhood Development (LEED-ND) was established as a methodology for the research due to the similarity in size and complexity involved in the project. The Growth Acceleration Program (PAC), aligned with the global guidelines for urban development and presented a discourse on sustainable development, however, it did not require that its projects have a certification with any Environmental certification [37]. In this way, the novelty of this research arises from the need to analyze the existing relationships, between urban development actions and sustainability, through LEED-ND indicators, in the proposals of the Brazilian government plan PAC2, as a means of verifying their effectiveness. Therefore, this work aimed to analyze the projects of the "Urbanization of Slums" sub-axis of PAC2 of the Morro do Encontro, still in the executive design phase; identify the empirical evidence of sustainability indicators in the selected projects, and compare the indicators extracted from the projects with the LEED-ND indicators. Once a project is finished and assessed, a political decision will determine its implementation.

This work aims to contribute to the historical survey of Brazilian urban development (understood as a possible means of urban management for the application of sustainable solutions), as well as to the discovery of the measurement of urban sustainability applied in a real project. The relevance of this study is represented by the following points: (1) the strong trend of urban growth, and mainly, the speed with which the slum areas in Rio de Janeiro increase; (2) the existence of a program with a general discourse on sustainability, but which did not require any sustainable certification for the urbanization of slum areas; (3) understanding that sustainable cities are long-term goals; (4) in the scientific dissemination of public actions that underpin sustainable urban development on an ongoing basis. This work could facilitate a better understanding of the existing relationships between urban development actions and sustainability based on a comparative analysis between sustainability indicators and project strategies.

## 2. Materials and Methods

This work has the purpose of contributing strategically in such a way that the knowledge can eventually be used in the solution of specific problems. It has the general descriptive purpose in search of maximum retraction of the characteristics of the selected project in a way to identify relationships with the study variables of the LEED-ND sustainability indicators [38]. In approaching the research methodology, data will be treated in a qualitative and quantitative way. In other words, the qualitative

refers to its subjective in the interpretation of the data whereas it is still quantitative, as it has an objective character applied for data analysis. The qualitative part refers to the identification of empirical evidence of sustainability indicators in the selected projects and the quantitative part refers to the comparison of the indicators extracted from the projects with the LEED-ND indicators [37]. The type of reasoning for the search for results is generalist, with specific observations to obtain general conclusions. Its technical procedures involve bibliographic research processes in scientific documents. This work illustrates a cross-sectional descriptive study of documentary analysis, within an objective of studying the project of the second step of the Growth Acceleration Program (PAC2) [39], in its sub-axis "Urbanization in Precarious Settlements" [40], with the transformation from a symbolic language of technical design to a textual and numerical language. It should be noted that the project is still in the executive design phase.

The proposed research method of this work is illustrated in Figure 1. It uses as a base example a project of the second stage of the Growth Acceleration Program (PAC2) and the LEED-ND certification process. The steps of the qualitative analysis included the presentation of the scenario diagnosis, the presentation of the collected sample and data collection, as presented in Figure 1. The steps of the qualitative analysis included the evaluation of the credit items (CI) and the mandatory items (MI) between the evidences of sustainability identified in the project, and, finally, the identification of potential sustainability items that can still be developed to meet resulting in certification.

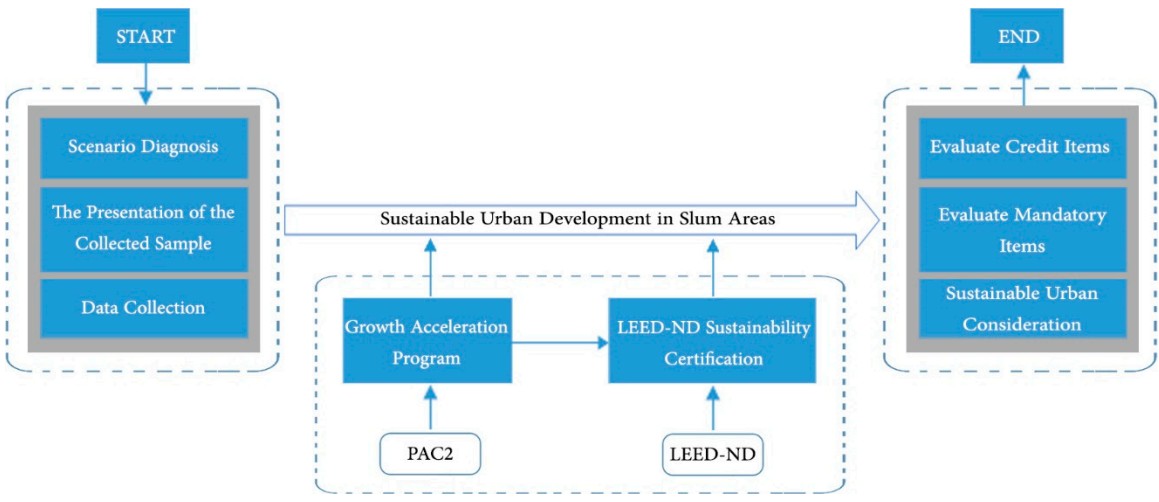

**Figure 1.** Proposed research method.

### 2.1. Scenario Diagnosis

In Brazil, between 1970 and 2010, the urban evolution rate was close to 28% [41]. Among the five Great Brazilian Regions in the Southeast, it was the one that maintained the prominent position, with an increase that almost made it urban in its entirety during the analyzed period. Southeast Brazil is the most urbanized region in the country, it is also the one with the highest population, despite occupying the penultimate position in territorial dimensions. Thus, almost half of the entire urban population in Brazil lives in the Southeast region [42]. Rio de Janeiro, the capital of the state that bears the same name, is located on the Southeast region of Brazil and is a 100% urban city. According to the IBGE, the 2010 demographic census pointed out that this city has a total population of 6,320,446 inhabitants, in an area of 1226.66 km$^2$, which totals a demographic density of 5154.68 inhabitants/km$^2$ [43].

Complexo do Lins is a large slum in Rio de Janeiro that is part of the neighborhood of Lins de Vasconcelos (a middle class neighborhood in the North Zone of the city), located in the administrative region of Greater Méier, on the slopes of the Massif da Tijuca Forest [44]. Figure 2 presents an overview of the slum area together with the delimitation of its 15 hills, where the white area determines the Morro do Encontro, the local of the case study of this work. With a total of 65,813 m$^2$, Morro do Encontro occupies the fourth largest territorial extension of the Priority Investment Project (PIP) of Lins

Complex [45,46]. The relief of the Lins Complex has large gaps, and the occupation levels of Morro do Encontro follows the same pattern and varies on average of 70 m (from 60 to 130 m), occupying one of the highest positions in the entire complex. The data are according to CONSORTIUM [47].

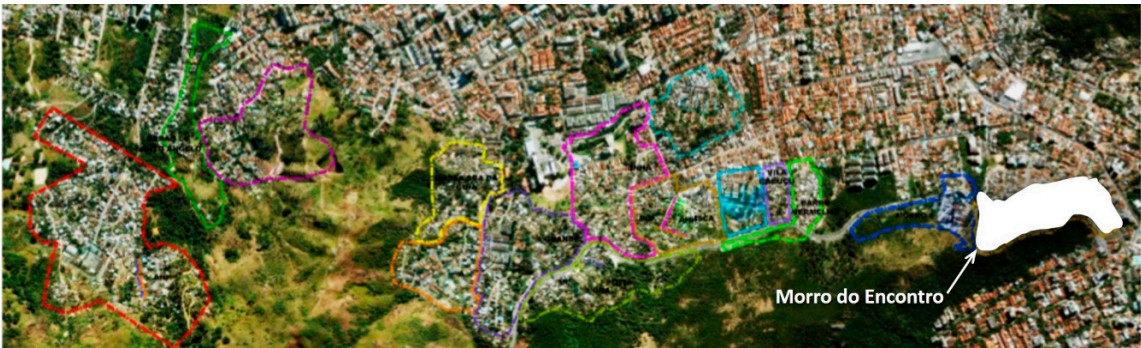

**Figure 2.** Scenario: general context (Lins Complex as seen from above) [45].

### 2.2. The Presentation of the Collected Sample

The case study example of this study consisted of the boards of the PAC2 executive project in Morro do Encontro, a project still in the executive design phase, available in the digital medium, from a project developed by a bidding consortium, and managed and supervised by the Public Works Company of the State of Rio de Janeiro (EMOP) [48]. The project boards comprised of technical drawings, detailed with dimensions, situation plan, an indication of the north, legend of materials, urban furniture, colors, outlines, and coordinates. The boards were developed in A1 format, in printed and digital media, with floor plans and cuts, with variable scales, being 1/1000, 1/500, 1/250, and 1/100. The sample selection was divided in four steps. The first step was to collect the set of Urbanism PAC2 projects, of Morro do Encontro. It is consisted of 79 boards. The second step was to select the ones in the executive design phase, making up a total of 36 boards. The third step was determined to include those that had already been approved by EMOP, totaling 36 boards. The fourth step excluded those whose drawings were of constructive detail. In this way, the final sample resulted in 12 boards. Each of the steps in this process are represented according to Figure 3.

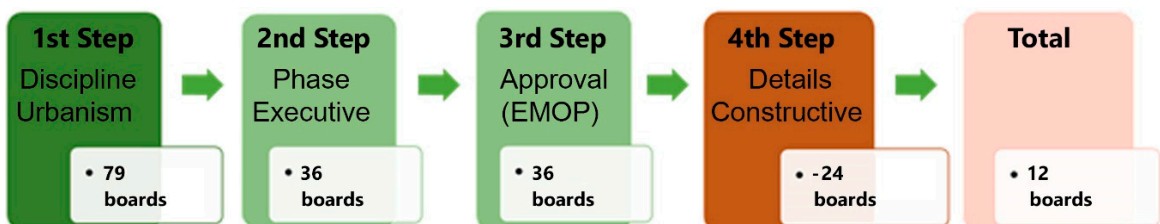

**Figure 3.** Sample selection steps.

### 2.3. Data Collection

Data collection was carried out in three main steps as highlighted in Figure 4. The first step of data collection consisted of the electronic extraction in full of the project present in the archives of the EMOP of the State of Rio de Janeiro. The Manual used for this analysis was LEED v4 for Neighborhood Development in its updated English version of April 5, 2016 [30]. The data was stored in an Excel for Windows spreadsheet with the following data found on each of the boards: name of the drawing; revision number; date; format; scale; unit of measurement; the presence of captions, presence of details, coordinates, presence of evidence, and general observations.

| 1st Step | 2nd Step | 3rd Step |
|---|---|---|
| Extraction of empirical evidence implicit in urban projects in the slum Morro de Encontro | Transcription of empirical evidence for LEED-ND indicators | Comparison between the indicators found in the projects with the LEED-ND indicators |

**Figure 4.** Data collection steps.

The second stage was composed of the empirical evidence raised in the previous stage and submitted to an analytical process for the designation of sustainability indicators, according to LEED-ND. The evidence found was transcribed in a subdivided manner in five thematic contexts and along the same lines as LEED-ND, being Intelligent Location and Connections; Neighborhood Design; Green Buildings and Infrastructure; Innovation and Design Process; and Regional Credits. The third step was the comparison, between the empirical evidence of sustainability and the LEED-ND indicators.

*2.4. Growth Acceleration Program*

The Growth Acceleration Program herein represents an initiative to produce sustainability applied through public policy. It consists of a Brazilian government program that, in dialogue with the private sphere, sought to balance the economy of the country by accelerating and sustaining growth. At the same time, it aims to reduce poverty and social inequality, preserve price stability and reduce State indebtedness, besides, to generate job opportunities and income. This program was sustained in two stages, which were known by the acronyms PAC and PAC2 [49]. The program has a diverse area of activity such as transport, energy, water, and electricity [39]. Over the years, its structure has evolved, differentiating from the first to the second stage; in both stages, urbanization in precarious settlements, slums, was contemplated. Therefore, when the activities of PAC in the slum areas are defined as Priority Investment Project (PIP), the Social and Urban Infrastructure works, which are under the management of the Ministry of Cities (MCIDADES), are the implementation of integrated housing actions, and sanitation and social inclusion. Meanwhile, operationalization had been carried out by Federal Savings Bank—CEF (Caixa Econômica Federal) [50]. Figure 5 illustrates the areas of intervention that are always very precarious.

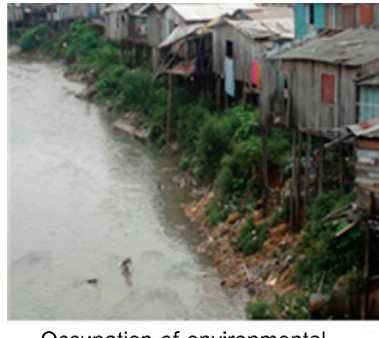

Occupation of environmental protection areas. Manaus / AM

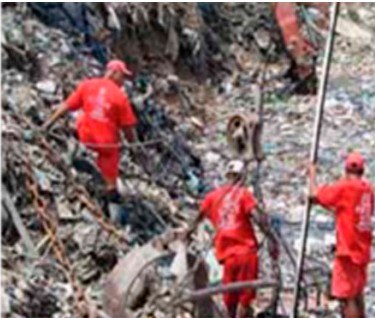

Recovery of Rio Beberibe. Recife PE

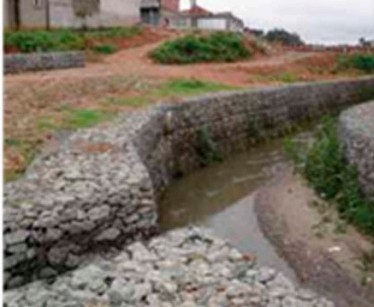

Billings / Guarapiranga stream margin recomposition. Sao Paulo-SP

**Figure 5.** PAC: intervention areas [36].

The second stage of the PAC began in 2011 and was scheduled to last until 2014, and emerged from the Federal Government's announcement of 29 March 2010 [50], Decree No. 7470 of 4 May 2011 [51]. It became known as PAC2 and maintained its strategic guidelines and the objective of overcoming the infrastructure bottlenecks in Brazil. The program was restructured into six complementary areas under the following names: Better City; Citizen Community; My home, my life; Water and Light

for All; and Transport and Energy [51]. The sub-axis of "Urbanization of Precarious Settlements", was understood in the urban centers' scenario, and had activities that foresaw housing construction, drainage, water supply, and sanitation and public lighting. It was just one of the 15 branches of the PAC2 Social and Urban Infrastructure axis [50]. In the context of the capital of Rio de Janeiro and its many precarious settlements, the urbanization project for five slum areas was contracted during the second phase of the PAC; the Rocinha Complex, the Tijuca Complex, the Jacarezinho Complex, the Mangueira Complex, and the Lins Complex were contemplated. For each of these slum areas, a bidding process was opened in which those companies, or a group of consortium companies that could meet the needs of the project development contract, could participate until their executive project phase. Meanwhile, the Rio de Janeiro Public Works Company (EMOP) was in charge of managing the contract [52].

### 2.5. LEED-ND Sustainability Certification

The application of the indicators can culminate in obtaining certificates, which will work as a tool to attest sustainability in certain aspects such as the certification model used as a research method in this work, the Leadership in Energy and Environmental Design (LEED®). This certification system emerged in 1993 and was designed by members of the United States Green Building Council (USGBC). However, its launch took place only in 1998 throughout the United States [53], while its introduction in Brazil occurred initially in the southern region of the country since 2005, around seven years after its launch in the United States [54]. The LEED tool became a model for measuring Green Design (Green/Ecological Design), which through accepted standards and methodologies are serving for the statistical analysis of sustainability [55,56]. The LEED system has already been applied in at least 150 countries [57]. Considering a world ranking, Brazil ranks 3rd among the countries with the largest number of enterprises with this certification [46], a fact that characterizes its good acceptance in this country. Assessing the application of sustainability in an urban environment, LEED developed a specific typology of its indicator system called Neighborhood Development (LEED-ND).

The concept of Sustainability has evolved over the years in the same way the systems of sustainability indicators have evolved. Thus, this work applies the most updated version of the LEED-ND system (LEED v4 for Neighborhood Development) [30]. The project must be framed in one of the subdivisions of LEED within a minimum score to receive certification [51]. The score can vary between 40 and 110 points, and the resulting levels can be *Certified, Silver, Gold, and Platinum* [47]. The score must be distributed according to the checklist of each typology. Each checklist is made up of thematic divisions into large groups. Finally, the groups are organized between prerequisite variable points (do not score) and credits (score differently), in which it is possible to identify an excerpt from the LEED-ND score sheet, about neighborhood development [38]. The LEED-ND typology has its points distributed in three main axes with items of prerequisites and credits (such as Intelligent Location and Connections, Neighborhood Design, and Green Infrastructure and Buildings), in addition to the credit items in Innovation and Design Process and Regional Credits [38].

The "Smart Site and Connections" has five mandatory items. These are smart site, species at risk and ecological communities, conservation in wetlands and water bodies, conservation of agricultural land, and the prevention of flood plains. In addition to these, there are nine other credit points, totaling 28 points [38]. The "Neighborhood Standard and Project" has three mandatory items. These are walkable streets, compact development, and a connected and open community. In addition to these, there are 15 additional credit items, totaling 41 points [38]. The "Green Infrastructure and Buildings" has four mandatory items. These are certified green buildings, minimum energy performance of the building, reduction of water use in the interior, and the prevention of pollution in the construction activity. In addition to these, there are 17 other credit points, totaling 31 points [38]. The "Innovation and Design Process" can consist of up to five points in innovation credits and one credit on the LEED accredited professional, totaling six points [38]. The "Regional Credits" can be composed of up to four points in four different items, both modalities have only items of credits [38]. Similarly to LEED types,

LEED-ND must be composed of between 44 and 110 points to receive one of the four certifications (*Certified, Silver, Gold or Platinum*).

The methodological steps adopted herein include the presentation of the Scenario Diagnosis, the Presentation of the Collected Sample, the Data Collection, the Treatment of the Collected Data, and finally the Presentation of the LEED-ND Indicators. At this level of the analysis, the Scenario Diagnosis consists of a general presentation of the Lins Complex Scenario [47], and later a Presentation of the Collected Sample of the PAC2 project [48]. The Data Collection presents the description of its three stages (extraction of empirical evidence from urban projects, transcription of empirical evidence for LEED-ND indicators, and comparison between project indicators versus LEED-ND indicators) which dealt with the extraction and transcription of the sustainability evidence found, the comparison between these stages and the LEED-ND sustainability indicators [38]. The next step consisted of the analysis and treatment of the collected data and finally, the last methodological step of the work consisted of presenting the sustainability indicators chosen for the development of this research.

## 3. Results

At this level of the analysis, the results from the methodological steps of this research were presented in the form of figures and tables. The comparison between empirical evidence of sustainability and LEED-ND indicators occurred through a transformation from the symbolic language from technical design to textual and numerical language. Table 2 illustrates the 26 evidences of sustainability extracted from the Morro do Encontro project.

**Table 2.** Morro do Encontro: evidence of sustainability identified in the project.

| | Evidence of Sustainability Extracted from the Project |
|---|---|
| 1 | Water infrastructure |
| 2 | Sewage infrastructure |
| 3 | Reforestation (Georio) |
| 4 | Reforestation (PAC) |
| 5 | Eco-limit |
| 6 | Community garden |
| 7 | Open spaces (Urban Oxygenators) |
| 8 | Habitat Conservation |
| 9 | Containment walls (Georio) |
| 10 | Containment walls (PAC) |
| 11 | Previously occupied location |
| 12 | Location in need of development |
| 13 | Pre-existing connectivity |
| 14 | Government program |
| 15 | Bus bays within the proposed limits |
| 16 | Staircase recovery |
| 17 | Removing homes in risk areas |
| 18 | New road with sidewalks |
| 19 | Recovery of sports areas |
| 20 | Restoration of leisure areas |
| 21 | Main access revitalization |
| 22 | Participatory research methodology |
| 23 | Research on secondary bases |
| 24 | Preservation of vegetation |
| 25 | Steep slope protection |
| 26 | Sediment control |

The correlations between the "Empirical evidence of sustainability implicit in the project object of this study, and the "LEED-ND sustainability indicators" were analyzed. According to the LEED-ND "Checklist", the evidence of the sustainability of a project is separated into mandatory items and credit items. The evidence found covered a total of 33% of mandatory items and 8% of credit items. In these

terms, the LEED-ND "Checklist" includes a possible total of 12 mandatory items and 110 points in credit items. In this way, the percentages previously presented are the result of the total attendance of four mandatory items and nine points in credit items, as shown in Figure 6.

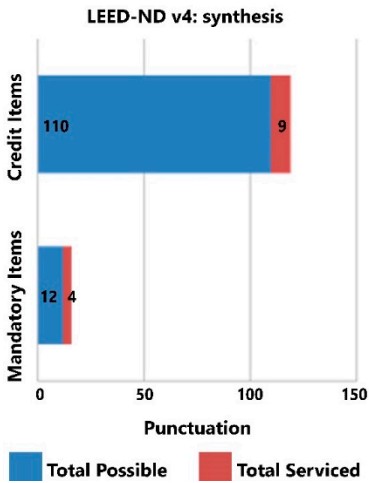

**Figure 6.** Numerical synthesis of the sustainability evidence found.

The output results pointed to the existence of attempts to apply sustainable solutions. However, these results are insufficient for certification by the LEED-ND indicator system, as the mandatory items were not fully met. However, for a more in-depth understanding of the sustainability of the project in question, the number of items served (mandatory items and credit items) by LEED-ND thematic contexts was extracted from the research. LEED-ND includes five thematic contexts, each with its checklist according to Table 3.

**Table 3.** Leadership in Energy and Environmental Design for Neighborhood Development (LEED-ND): thematic contexts.

| Thematic Contexts of Neighborhood Sustainability by LEED-ND | | |
|---|---|---|
| | **Mandatory Items** | **Credit Items** |
| Smart location and nexus | 5 | 28 |
| Neighborhood pattern and design | 3 | 41 |
| Green buildings and infrastructure | 4 | 31 |
| Innovation and design process | 0 | 6 |
| Regional priority credits | 0 | 4 |
| Total | 12 | 110 |

Each of the thematic contexts were analyzed separately and based on the results, the minimum attendance of the items was checked if verified or not. Table 4 identifies the presence of mandatory items in the project, confirming that without their application, LEED certification would be impossible. Through the analysis of the project, it was seen that the mandatory items were found in only three of the five thematic groups. These are Intelligent location and bond, Neighborhood pattern and design, and Green infrastructure and buildings. The total number of mandatory items in this stage was 12, as shown in Table 4. According to the exposed results in Table 4, it is observed that:

- "Smart location and bond" is a thematic group composed of five mandatory items, of which 100% have met the LEED-ND requirements.
- "Standard and neighborhood project" is a thematic group composed of three mandatory items, none of which met the LEED-ND requirements.

- "Infrastructure and green buildings" is a thematic group composed of four mandatory items, none of which met the LEED-ND requirements.
- Therefore, it can be said that the complete analysis of all checklists indicated the existence of a total of 12 mandatory items, of which only four were met, a total of 33%.

**Table 4.** Correlation of sustainability: mandatory items.

| Mandatory Items LEED-ND | | | | | | | | | | | | | | | | | |
|---|---|---|---|---|---|---|---|---|---|---|---|---|---|---|---|---|---|
| Evidence of sustainability | 1 | 2 | 3 | 4 | 5 | 6 | 7 | 8 | 9 | 10 | 13 | 16 | 18 | 19 | 20 | 24 | 26 |
| Smart location and nexus | | | | | | | | | | | | | | | | | |
| Smart location | x | x | | | | | | | | | x | x | | | | | |
| At-risk species and ecological communities | | | x | x | x | | | | | | | | | | | | |
| Conservation of floodplain and water bodies | | | | x | | | | | | | | | | | | | |
| Conservation of arable land | | | | | x | | | | | | | | | | | | |
| Flood quota spacing | | | | | | x | x | x | x | x | | | x | x | x | x | |
| Neighborhood pattern and design | | | | | | | | | | | | | | | | | |
| Walkable streets | | | | | | | | | | | | | x | | | | |
| Compact development | | | | | | | | | | | | | | | | | |
| Connected and open community | | | | | | | | | | | | | | | | | |
| Green buildings and infrastructure | | | | | | | | | | | | | | | | | |
| Certified buildings | | | | | | | | | | | | | | | | | |
| Minimum energy efficiency in buildings | | | | | | | | | | | | | | | | | |
| Minimum water efficiency in buildings | | | | | | | | | | | | | | | | | |
| Pollution prevention in construction activity | | | | | | | | | | | | | | | | | x |

Regarding the comparison between the empirical evidence of sustainability in the Lins PAC2 Complex projects and the LEED-ND credit items, the following Table 5 was produced: "Correlation of sustainability: credit items". LEED-ND has 47 credit items with different weights, which can add up to 110 points, and are presented in all thematic groups of this system. That are Smart location and bond, Neighborhood pattern and design, Green buildings and infrastructure, Innovation and design process, and Regional priority credits.

**Table 5.** Correlation of sustainability: credit items.

| Empirical Evidence of Sustainability Implicit in the Project | | | | | | | | | | | | | | | | | |
|---|---|---|---|---|---|---|---|---|---|---|---|---|---|---|---|---|---|
| Evidence of sustainability | 2 | 3 | 4 | 5 | 8 | 9 | 10 | 11 | 12 | 13 | 14 | 15 | 17 | 21 | 22 | 23 | 25 |
| Smart location and nexus | | | | | | | | | | | | | | | | | |
| Preferred location | | | | | | | | x | x | x | | | | | | | |
| Re-urbanization of contaminated areas | | | | | | | | | | | x | | | | | | |
| Quality access and transport | | | | | | | | | | | | x | | | | | |
| Steep slope protection | | | | | x | x | x | | | | | | x | | | | x |
| A project of land for the conservation of habitat or wetlands and bodies of water | | x | x | x | | | | | | | | | | | | | |
| Other credit items | | | | | | | | | | | | | | | | | |
| Neighborhood pattern and design | | | | | | | | | | | | | | | | | |
| Access to civic and public spaces | | | | | | | | | | | | | | x | | | |
| Community outreach and involvement | | | | | | | | | | | | | | | x | x | |
| Other credit items | | | | | | | | | | | | | | | | | |
| Green buildings and infrastructure | | | | | | | | | | | | | | | | | |
| Reduce terrain disturbance | | | | | | | x | | | | | | | | | | |
| Other credit items | | | | | | | | | | | | | | | | | |
| Innovation and design process | | | | | | | | | | | | | | | | | |
| Minimum water efficiency in buildings | | | | | | | | | | | | | | | | | |
| Pollution prevention in construction activity | | | | | | | | | | | | | | | | | |
| Regional priority credits | | | | | | | | | | | | | | | | | |
| Regional Credit | | | | | | | | | | | | | | | | | |

Therefore, it can be said that the complete analysis of all checklists showed a total of 8% of the 110 points possible credit items; only nine points were met. To present the number of sustainability items found in the project, Table 6 was elaborated that discriminates each of its items. The items were organized based on the total number of possible mandatory items and the total number of possible credit items for certification. The number of items attended, which may be mandatory or credits, and the number of points according to the type and quantity of credit items found are also presented. The results between the number of credit items can be varied from their score obtained since each item has an assessment weight that is different from the others. Finally, the result of meeting the thematic context is presented in the last column of Table 6.

**Table 6.** Number of mandatory items (MI), number of items attended (IA), number of max credit items for certification (CI), and LEED-ND score (LS).

| Thematic Context LEED-ND | N° of Mandatory Items | N° of Credit and Score Items | Did You Attend the Thematic Context? |
|---|---|---|---|
| Smart location and nexus | MI = 05 IA = 04 | CI = 28 points IA = 05/09 LS = 07 points | Yes |
| Neighborhood pattern and design | MI = 03 IA = 0 | CI = 41 points IA = 01/15 LS = 01 points | No |
| Green buildings and infrastructure | MI = 04 IA = 0 | CI = 31 points IA = 01/17 LS = 01 points | No |
| Innovation and design process | MI = 0 IA = 0 | CI = 06 points IA = 00/02 LS = 00 points | No |
| Regional priority credits | MI = 0 IA = 0 | CI = 04 points IA = 00/04 LS = 00 points | No |
| **Total** | MI =12 IA = 04 | CI = 110 points IA = 07 LS = 09 points | No |

As can be seen in Table 6, the thematic context "Smart location and nexus" was the only one to meet 100% of its mandatory items, in addition to registering seven points in credit items (out of a total of 28 points). Hence, this context was the only one who contemplated its thematic context as "attended to". Although this survey included a total of 11 items attended (four being mandatory and seven being credits), LEED-ND certification would be impossible. As already mentioned, most of the mandatory items were not met. These results indicate the existence of attempts in the application of sustainable solutions that, however, would not be sufficient for any certifications by the LEED-ND sustainability indicator system since it has not met the 100% of mandatory items. Likewise, there was no minimum attendance of the 40 points required for certification, a fact that reinforces the impossibility of sustainable LEED-ND certification. Besides, the 40 points of their minimum credit items for a LEED-ND certification were not considered. Finally, after analyzing the entire project, a total of 47 sustainability items were identified, which can still be studied and developed to meet LEED-ND and resulting in certification, as shown in Table 7.

**Table 7.** The 47 potential sustainability items.

**Smart Location and Links**
Credit items (04/09):
- Bicycle facilities
- Proximities between home and work
- Restoration of habitat or wetlands and water bodies
- Long-term conservation management of habitat or wetlands and water bodies

**Neighborhood Pattern and Design**
Mandatory items (03/03):
- Walkable streets
- Compact development
- Connected and open community

**Table 7.** *Cont.*

| Credit items (14/15): |
| :--- |

- Walkable streets
- Compact development
- Mixed-use neighborhoods
- Residential typologies and accessible values
- Reduction of the parking projection area
- Connected and open community
- Transit facilities
- Transport demand management
- Access to civic and public spaces
- Access to leisure facilities
- Visibility and universal design
- Local food production
- Shaded and shaded urban landscape
- Neighborhood schools

**Green Buildings and Infrastructure**
Mandatory items (04/04):

- Certified green building
- Minimum energy performance of the building
- Reduction of water use in the interior
- Pollution prevention in construction activity

Credit items (16/17):

- Certified green buildings
- Optimize the energy performance of the building
- Reduced indoor water use
- Reduction of water use abroad
- Building reuse
- Preservation of historical resources and adaptive reuse
- Stormwater management
- Reduction of heat islands
- District cold water and heating plant
- Energy efficiency of infrastructure
- Solid waste management
- reduction of light pollution

**Innovation and Design Process**
Credit items (02/02):

- Innovation
- LEED accredited professional

**Regional Priority Credits**
Credit items (04/04):

- Regional priority credit: defined region
- Regional priority credit: defined region
- Regional priority credit: defined region
- Regional priority credit: defined region

## 4. Conclusions

This work aimed to analyze a real urban project in the scenario of Rio de Janeiro based on the LEED-ND indicator system. The innovation of this research arises from the need to analyze the existing relationships, between urban development actions and sustainability, through LEED-ND indicators, in the proposals of the Brazilian government plan PAC2, as a means of verifying their effectiveness. This work could help make a better understanding of existing relationships between urban development actions and sustainability based on a comparative analysis between sustainability indicators and project strategies.

1. The themes between urban development and sustainability were analyzed before the current scenario in the context of slum areas in the city of Rio de Janeiro. The correlations between a recent PAC2 project in the executive design phase (in its sub-axis "Urbanization of Slums") were considered with the application of a system of urban sustainability indicators of national and international recognition, the LEED-ND. This sustainability indicator (LEED-ND) was selected as a specific type of LEED that deals with *Neighborhood Development*".

2. During the analysis of 12 boards of the Morro do Encontro Executive Urbanism Project, a total of 11 sustainability items were divided between mandatory and credit items. Finally, after applying the weights of each credit item, the final result was concluded in nine points; as for mandatory items, these points were not fully met. Therefore, this issue demonstrates the reasons why the analyzed project could not be certified by the LEED-ND system. However, the final balance of the survey showed a total of 47 sustainability items with the potential for development to adapt to the LEED-ND sustainability criteria. The following issues were then considered; (1) the project has not yet been executed (due to the end of the Program); (2) the project has the potential to be adjusted according to the LEED-ND criteria; (3) the study scenario lacks emergency improvements to bring dignity and quality of life to its inhabitants. Therefore, it was evident that there were attempts to apply sustainable solutions to the project, however, it was insufficient. Given all the issues presented, the importance and complexity of the theme addressed are highlighted.

3. These results are expected to support more inclusive development actions. Therefore, the imperative in the search for the study, development and dissemination of a more sustainable and humane mentality is reinforced. As a suggestion for future work, the analysis of these projects based on other systems of recognized sustainability indicators is indicated. In this way, it can be said that all the research objectives were met and included: the analysis of the projects of the sub-axis "Slum Urbanization" of PAC2 of the "Morro do Encontro favela"; the identification of empirical evidence of sustainability indicators in the selected projects; comparing the indicators extracted from the projects with the LEED-ND indicators.

4. Despite the contemporaneity of the theme addressed and the strong presence of LEED in Brazil, the lack of material available for LEED-ND in the Portuguese language was among the difficulties faced by the research. Therefore, it can be said that given the need for adequate planning to accompany the projections of urban growth as a trend, and the need to sustain life on Earth that has existed for decades, a very complex, current, and evidently social valuable issue is faced. Additionally, it became clear that attempts at sustainable solutions have been applied in several places for years, yet unsustainable human settlements, subnormal agglomerations (such as slums) persist, reinforcing limitations on the right to the city.

**Author Contributions:** Conceptualization, A.C.H.d.S. and E.V.; Data curation, M.K.N.; Formal analysis, A.C.H.d.S. and A.W.A.H.; Investigation, A.C.H.d.S., A.H. and E.V.; Methodology, A.C.H.d.S. and E.V.; Resources, A.C.H.d.S.; Software, M.K.N.; Supervision, A.H. and E.V.; Validation, A.W.A.H. and E.V.; Writing—original draft, A.C.H.d.S. and M.K.N.; Writing—review and editing, M.K.N., A.W.A.H., A.H. and E.V. All authors have read and agreed to the published version of the manuscript.

**Funding:** This research was funded by COPPETEC Research Project number [17902]. Additionally, the financial support from CNE FAPERJ 2019-E-26/202.568/2019 (245653) Fundação de Amparo à Pesquisa do Estado do Rio de Janeiro, and CNPq (Brazilian National Council for Scientific and Technological Development) grant number [307084/2015-9].

**Acknowledgments:** The authors want to thank Departamento de Construção Civil, Poli-UFRJ, in facilitating equipment installation and resources for the development of this project.

**Conflicts of Interest:** The authors declare no conflict of interest.

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
