# Peer review of "Sustainable Urban Development in Slum Areas in the City of Rio de Janeiro Based on LEED-ND Indicators"

_buildings, doi:10.3390/buildings10070116_

Round 1

Reviewer 1 Report

The paper clearly returns the research path undertaken by the authors. Objectives and methodology are explained explicitly. The results support the initial hypotheses. Empirical data and case studies are sufficient.

Author Response

Ms. Raluca Istoan Assistant Editor Journal Buildings June 11, 2020

Dear Ms. Raluca Istoan,

Manuscript ID: buildings-807951

Please find attached a revised version of our manuscript, “Sustainable Urban Development in Slum Areas in the City of Rio de Janeiro based on LEED-ND Indicators”, submitted as an Article in Journal Buildings, after we perform the requested reviews.

The comments on the review form were highly insightful and enabled us to greatly improve the quality of our manuscript. In the following pages are our point-by-point responses to each of the comments of the editor and reviewers. Revisions in the text, below, are shown using the yellow highlight for additions. In accordance with the suggestions, we presented the goals and objectives in a separate subsection in the Introduction Section, enhanced the Materials and Methods Section as well as the Results Section. Hence, Figure 2 in the previous submission was removed, while Table 7 has been added to the revised manuscript. Besides, a Graphical Abstract has been submitted in this round of revision.

We hope that the revisions in the manuscript and our accompanying responses will be sufficient to make our manuscript suitable for publication in Journal Buildings.

We shall look forward to hearing from you at your earliest convenience.
Yours sincerely,
Ana Carolina Siqueira, Mohammad K Najjar, Ahmed W A Hammad, Assed Haddad, and Elaine Vazquez

Reviewer 2 Report

  1. This is more a technical note than a scientific article. The authors write that "The novelty of this study is to evaluate the indicators of LEED-ND and verify whether the requirements established by this method are present in a real urban development project in slum areas in Rio de Janeiro." The authors do not solve the research problem, they only check the content of the project - they evaluate the project in accordance with the adopted indicators. Authors should indicate the contribution of this manuscript to science.
  2. Reading the article, we do not know whether the evaluated project is implemented, it is in the implementation phase or it will be implemented in the future. The explanation is only in paragraph 5. Conclusions - the authors write that the project has not yet been executed. This should be described in the abstract and paragraph 2 Materials and Methods.

The question arises whether the project will be implemented in the future? If not, what should this assessment give? Its results should serve to improve the project before implementation. Is the analyzed project likely to move to the implementation stage?

  1. In the conclusions, the authors write "the final balance of the survey showed a total of 47 sustainability items with the potential for development to adapt to the LEED-ND sustainability criteria". What about the missing mandatory items?

The authors also write "As a suggestion for future work, the analysis of these projects based on other systems of recognized sustainability indicators is indicated". Question: why didn't they do it? Maybe they should compare the results obtained here with the results based on other systems.

  1. Change the organization of the text. Paragraphs 5.1. Scenario Diagnosis, 5.2. The Presentation of the Collected Sample, 5.3. Data Collection should be moved to paragraph 2 Materials and Methods. They describe the research method used. Only results obtained should be discussed in the Results paragraph. The results are presented in paragraph 5.4. Treatment of the Collected Data. In addition, paragraph 2 Materials and Methods should be moved after paragraph 4. LEED-ND Sustainability Certification
  2. Each abbreviation should be explained the first time it is used - for example, in the summary, the abbreviations CI and MI
  3. Figure 2 is unnecessary, the same is in the text above. It adds nothing to the manuscript.
  4. Line 170 - the beginning of the sentence is missing.
  5. Figure 3 - there are two areas marked by the yellow dotted line. The figure is difficult to read. Colorful lines delimiting individual areas should be clearer.
  6. Table’s 6 title - "Number of Items Attended (IA), Number of Mandatory Items (MI)" should be "Number of Mandatory Items (MI), Number of Items Attended (IA)" because MI first appears in the table.

Author Response

(The authors gave the same response as above.)

Reviewer 3 Report

Some suggestions to improve the quality of the article:

  • There is no Literature Review, or State of Art, to clarify what was done regarding the sustainability of Rio de Janeiro, or other cities in Brazil. Which were the results and how were they applied?
  • Why is the title referring to Brazil, as long as the study targets only one city? Please revise the title.
  • How can the slums become part of the city? Do the regulations allow? Don`t they need approvals from the Municipality? Please explain.
  • Please provide more information on the qualitative-quantitative approach of your research.
  • A scheme of the steps you followed would be more visible for the readers for a better understanding. And, of course, detailing exactly what you did.
  • Can you provide specific updated data regarding the slums: number of people, area, level of income, facilities, degree of health, and others?
  • I don`t understand what is the purpose of the research. Did the authors take the data from the statistics? If this is the case, which is their contribution? What year are they referring at? Which is the sample they have studied? All of the slumps, or a specific area?
  • Which are the hypotheses of the research? How can the authors prove the validity and reliability of the results?
  • The references are outdated.

Author Response

(The authors gave the same response as above.)

Round 2

Reviewer 2 Report

I think it's much better now. First of all, the purpose of the work was clearly defined.

Author Response

Responses to the comments of Reviewer 2:

Reviewer 2:

I think it's much better now. First of all, the purpose of the work was clearly defined.

Response: Thanks for your time in reviewing our manuscript.

Reviewer 3 Report

The authors made the requested changes. However, the method used for the research remains unexplained. Only tables with indicators and results are not supported.

Author Response

Responses to the comments of Reviewer 3:

Reviewer 3:

The authors made the requested changes.

attached in the file bellow

Round 3

Reviewer 3 Report

The authors made all the changes.